

# Do goats recognise humans cross-modally?

Marianne A. Mason[1,2], Stuart Semple[1], Harry H. Marshall[1,3] and Alan G. McElligott[4,5]

[1] School of Life and Health Sciences, University of Roehampton, London, United Kingdom
[2] Department of Animal and Veterinary Sciences, Aarhus University, Tjele, Denmark
[3] RSPB Centre for Conservation Science, Cambridge, United Kingdom
[4] Department of Infectious Diseases and Public Health, Jockey Club College of Veterinary Medicine and Life Sciences, City University of Hong Kong, Hong Kong, Hong Kong SAR, China
[5] Centre for Animal Health and Welfare, Jockey Club College of Veterinary Medicine and Life Sciences, City University of Hong Kong, Hong Kong, Hong Kong SAR, China

Corresponding authors
Marianne A. Mason,
masonm@anivet.au.dk
Alan G. McElligott,
alan.mcelligott@cityu.edu.hk

## ABSTRACT

Recognition plays a key role in the social lives of gregarious species, enabling animals to distinguish among social partners and tailor their behaviour accordingly. As domesticated animals regularly interact with humans, as well as members of their own species, we might expect mechanisms used to discriminate between conspecifics to also apply to humans. Given that goats can combine visual and vocal cues to recognise one another, we investigated whether this cross-modal recognition extends to discriminating among familiar humans. We presented 26 goats (17 males and nine females) with facial photographs of familiar people and two repeated playbacks of a voice, either congruent (from the same person) or incongruent with that photograph (from a different person). When cues were incongruent, violating their expectations, we expected goats to show changes in physiological parameters and moreover, respond faster and for longer after playbacks. Accordingly, heart rate decreased as the playback sequence progressed, but only when the face and voice presented were incongruent. Heart rate variability was also affected by congruency, but we were unable to determine precisely where differences lay. However, goats showed no changes in time taken to respond, or how long they responded for (our primary variables of interest). We also found evidence to suggest that shifts in cardiac responses may not have been robust. Although our findings could imply that goats had successfully perceived differences in congruency between the visual and vocal identity information presented, further evidence is needed to determine whether they are capable of cross-modal recognition of humans.

## INTRODUCTION

Recognition forms a foundation for complex social behaviour, enabling animals to discriminate among social partners (*e.g.*, between mates, kin and competitors) and tailor their behaviour accordingly (*Tibbetts & Dale, 2007*; *Wiley, 2013*; *Yorzinski, 2017*).[1] For

[1]Portions of this text were previously published as part of a preprint: https://doi.org/10.1101/2023.08.04.551944.

domesticated animals, close-contact husbandry tasks (*e.g.*, feeding, cleaning and health-checks) and repeated interactions (a prerequisite for recognition) with humans are a frequent occurrence. Through these interactions, dyadic social relationships can develop, requiring participants to recall outcomes of previous interactions with one another to anticipate the other party's future behaviour (*Hemsworth & Coleman, 2011*). This proximity to humans over thousands of generations has made the domestic environment a unique setting for the development of interspecific communication, potentially fostering more cognitively demanding forms of perception of our cues (*Avarguès-Weber, Dawson & Chittka, 2013*; *MacHugh, Larson & Orlando, 2017*).

Dogs (*Canis lupus familiaris*), cats (*Felis catus*), and horses (*Equus caballus*) can combine cues in two sensory modalities (visual and vocal) not only to recognise members of their own species, but also familiar humans (*Adachi, Kuwahata & Fujita, 2007*; *Proops, McComb & Reby, 2009*; *Taylor, Reby & McComb, 2011*; *Proops & McComb, 2012*; *Takagi et al., 2019*). Known as cross-modal recognition, such an ability infers the existence of a mental representation, for a conspecific or person, which can be used to compare against available cues (a prerequisite of individual recognition: *Proops, McComb & Reby, 2009*). This ability may enable these companion species to discriminate among people with greater accuracy and would be especially advantageous when cues in a particular modality are attenuated or unavailable (*Ratcliffe, Taylor & Reby, 2016*). For example, visual features become more difficult to discern over distance or when hidden by intervening objects (*e.g.*, trees and fences). Under these conditions, we may expect an animal to rely more on what they can feel, smell or hear over what they can see.

Humans typically invest less time interacting with individual livestock compared to companion animals like cats and dogs which share their homes, or with animals such as horses, with whom they form close working relationships. Although the less individualised relationships we form with farm animals may not be as favourable for the development of more sophisticated interspecific communication abilities, these animals have already been shown to have an impressive repertoire of social skills to call upon when interacting with humans (for review: *Jardat & Lansade, 2021*). Comparatively little is known about complex recognition of humans in livestock, although goats (*Capra hircus*) appear to use cross-modal recognition to distinguish among conspecific social partners (*Pitcher et al., 2017*).

Goats were among the first livestock species to be domesticated, approximately 10,500 years ago (*MacHugh & Bradley, 2001*; *MacHugh, Larson & Orlando, 2017*). Although goats were primarily domesticated for meat, milk and hair products (*MacHugh & Bradley, 2001*; but see pack goats: *Sutliff, 2019*), during early domestication, this species appears to have undergone strong selection for tameness (*Dou et al., 2023*); a process that has been identified as being pivotal for the development of more advanced social cognition of human cues (*Hare et al., 2005*). Indeed, goats have been shown to read a variety of human cues, from attentional cues (*Nawroth, von Borell & Langbein, 2015*; *Nawroth, Brett & McElligott, 2016*; *Nawroth, von Borell & Langbein, 2016*; *Nawroth & McElligott, 2017*) and facial expressions (frowning from smiling: *Nawroth et al., 2018*) to communicative

gestures (pointing and tapping: *Kaminski et al., 2005*; *Nawroth, von Borell & Langbein, 2015*; *Nawroth, Martin & McElligott, 2020*; review: *Mason et al., 2024*).

Cue use in goats when recognising conspecific social partners spans multiple sensory modalities. Goats are farsighted, have poor depth perception and compared to humans, perceive a more limited range of colours (review, *Adamczyk et al., 2015*). However, they have a wider visual range and still use vision in a variety of contexts, including social recognition with coat colour appearing important for kids to recognise their mothers (*Ruiz-Miranda, 1993*). Vocal cues are also important for early recognition between mothers and offspring (*Briefer & McElligott, 2011*; *Perroux, McElligott & Briefer, 2022*), with mothers responding to their own kids' calls over those of other familiar kids up to 13 months post-weaning (*Briefer, Padilla de la Torre & McElligott, 2012*). Goats can perceive a broad range of frequencies (60 Hz–40 kHz; *Adamczyk et al., 2015*), and moreover, can combine vocal with visual cues to discriminate between conspecific social partners. When presented with a pen mate, a less familiar herd member and a call from one of the pair, goats were able to match playbacks to the original caller, turning towards them accordingly (*Pitcher et al., 2017*). However, despite the extensive research into how goats recognise conspecifics, no investigation to date has explored cues that they might use to recognise their next most important social partners, humans.

To investigate whether goats can recognise familiar people, we presented subjects with a facial photograph and a vocal playback, which were either from the same person (so were congruent) or from different people (were incongruent). Images were used instead of their real-life counterparts to prevent unintentionally cuing the goats (*Samhita & Gross, 2013*) and to restrict cues available to the visual modality. Further, goats have been previously shown as capable of discriminating details present in both black and white (*Nawroth et al., 2018*) and colour photographs as used here (*Bellegarde et al., 2017*). Still images have been used in similar congruency paradigms for other species, and following these investigations (*e.g.*, *Adachi, Kuwahata & Fujita, 2007*; *Takagi et al., 2019*), we predicted that if goats can recognise humans cross-modally, they would respond faster and for longer when visual and vocal cues were incongruent, reflecting a violation of their expectations. We also predicted that goats may exhibit a faster heart rate and lower heart rate variability in the incongruent, compared to the congruent condition (associated with heightened physiological arousal: *Briefer, Tettamanti & McElligott, 2015*; *Baciadonna, Briefer & McElligott, 2020*). Increases in heart rate for example, have been observed when human participants are presented with a word in a font colour incongruent with its meaning (*e.g.*, the word 'red' in blue ink, *i.e.*, the Stroop test: *e.g.*, *Boutcher & Boutcher, 2006*). Alternatively, the novelty of the coupling between human visual and vocal stimuli presented per se, or how these pairings contradict goat expectations may elicit a decrease in heart rate (and increase in heart rate variability), potentially enabling goats to better process external stimuli (*Vila et al., 2007*; *Bradley, 2009*; *Skora, Livermore & Roelofs, 2022*). In carrying out this research, we aimed to determine whether goat ability to develop cognitive representations for known individuals, a building block of individual recognition, extends beyond their own species.

## MATERIALS AND METHODS

### Ethics statement

Our stimuli collection from human participants and all experimental procedures were approved by the University of Roehampton Life Sciences Ethics Committee (Ref. LSC 19/ 280), with the latter being in line with ASAB guidelines for the use of animals in research (*ASAB/ABS, 2020*). The feasibility and adequacy of protocols we used to address our research questions were additionally approved by a Life Sciences Research Student Review Board prior to onset of testing (protocols not externally registered). All tests were non-invasive and lasted a maximum of seven minutes per subject for each trial. No animals were euthanised and all were released into a large outdoor paddock to join the rest of the herd following testing. We collected signed consent forms from human participants who provided stimuli for the current experiment.

### Study site & sample population

We conducted experiments between 8th September–19th October 2020 and 19th May–16th July 2021 at Buttercups Sanctuary for Goats (http://www.buttercups.org.uk/) in Kent, UK (51°13′15.7″N 0°33′05.1″E). During these periods, the sanctuary was open to visitors who were freely able to approach and interact with the goats, including opportunities to feed them. Goats had daily access to a large outdoor area (approximately 3.5–4 acres) and were kept individually or in small groups within a large stable complex at night (mean pen size = 3.5 m$^2$). Throughout the day, animals had *ad libitum* access to hay, grass and water, and were supplemented with commercial concentrate according to age and condition. Subjects came from a variety of backgrounds having been removed from their owners due to mistreatment or when owners could no longer care for them. We selected a target sample size based on a power analysis of our results from the first year of testing (2020; using the response, time spent looking after the start of each voice playback) and on sample sizes of similar studies (*e.g.*, *Adachi, Kuwahata & Fujita, 2007*; *Proops, McComb & Reby, 2009*). Our final sample comprised 26 adult rescue goats (17 castrated males and nine intact females) that were well habituated to human proximity, were of various breeds and ages, and had resided at the sanctuary for over eight months (for detailed subject information, see Table S1).

### Stimuli collection & preparation

We selected goats that had been described by a particular caretaker, volunteer or frequent visitor (from whom stimuli were collected for the current experiment) as being particularly responsive to them. We assumed the reported preferential attention given to a particular person (hereafter known as a preferred person) would indicate goats were more familiar with their individual characteristics. We always tested goats with stimuli from one preferred person and one gender-matched, non-preferred caretaker to avoid potential reliance on gender as a cue and with the aim of ensuring subjects were familiar with all human stimuli presented.

In 2020, we collected photographs and voice samples from two male and three female caretakers. Photographs and voice samples for each person were collected in a single session

at a two-meter distance, outdoors (as stimuli would normally be experienced by goats) and at the same location and time of day. In 2021, we collected stimuli from five additional volunteers and frequent visitors to the sanctuary and one familiar staff member not involved with carrying out husbandry procedures (five women and one man) who had developed bonds with individual goats. For these, we took photographs and voice samples based on availability, meaning similarity in time of day could not be maintained, and location was sometimes shifted based on prevailing light conditions (otherwise conditions were as described above). Goats tested in 2020 were subject to photographic and vocal stimuli from a pair of caretakers, but in 2021, one of the two people goats experienced stimuli from (the preferred person) was not responsible for carrying out husbandry procedures (*e.g.*, hoof trimming), which goats likely perceived as aversive. Caretakers were also involved in feeding the goats, occasionally providing favoured food items, with the volunteers and frequent visitors regularly provisioning preferred food more specifically to particular goats. This could have affected the nuance of the relationship with their preferred person for goats tested in 2020 compared to those tested in 2021 (the year a goat was tested was controlled for during statistical analysis).

To collect visual stimuli, we asked each person to maintain a neutral expression and face the camera (Panasonic Lumix DMC-FZ45) before several frontal photographs of their head and shoulders were taken against a white background. Photographs were later edited so only the head and neck were visible, processed to improve clarity and brightness, before being blown up to A3 landscape size (slightly larger than natural head-size).

While collecting vocal recordings, we asked each person to say the phrase: "Hey, look over here", several times in a manner they would normally use to address the goats. Speech was intonated, and we avoided using goats' names and other potentially salient words (*e.g.*, food-related vocabulary) to test whether potential recognition can be generalised based on vocal features, rather than being restricted to specific familiar words or phrases (*Kriengwatana, Escudero & Ten Cate, 2015*). Voice samples were recorded using a Sennheiser MKH 416 P48 directional microphone and a Marantz PMD-661 digital recorder (sampling rate: 48 kHz, with an amplitude resolution of 16 bits in WAV format). We selected the clearest voice sample (mean sample length $\pm$ SD = 1.28 s $\pm$ 0.35) with the lowest background noise and shifted the mean amplitude to 70dB to ensure consistency between samples and compiled these into a playback sequence using Praat v.6.1 (*Boersma & Weenink, 2019*). Playback sequences comprised five seconds of silence before the first voice playback, followed by 10 s of silence, a repeat of that playback, and 30 s of silence. The 10 s following initiation of each of the two playbacks in a sequence will be hereafter referred to as the 'response period.'

## Experimental enclosure

The experimental enclosure was constructed out of opaque metal agricultural fencing and barred metal hurdles in a large outdoor paddock that goats had ready access to throughout the day. The enclosure was divided into five sections (Figs. 1; S1). Goats entered through the preparation pen where they were equipped with a heart rate monitor before experiments. Trials took place in the experimental arena. A corner of the arena (the holding pen) was

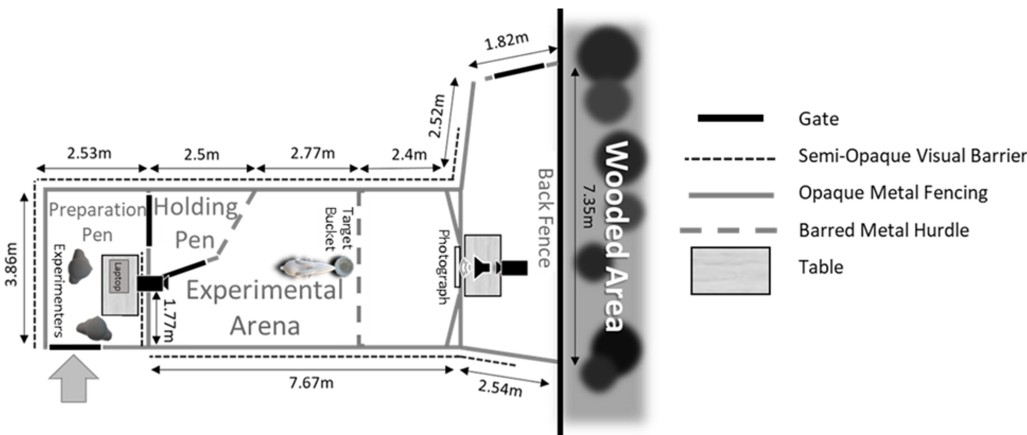

**Figure 1** **A schematic of the experimental enclosure used in 2020.** Goats entered the enclosure through the preparation pen where they were fitted with a heart rate monitor, before being moved into the experimental arena for training and experimental trials.

sectioned off with hurdles and used both for training (see 'Training Phase') and to prevent subjects from having visual access to experimenters during trials. We erected a semi-opaque green barrier around the fencing's edge to prevent subjects from being able to see inside the preparation pen and outside the arena to reduce distractions and unintentional cuing from experimenters, visitors and other goats. Two Sony CX240E video cameras (frame rate = 25 FPS) were positioned at the front and back of the arena. The camera at the front was hidden under camouflage netting to prevent subjects attributing it as the sound source. We placed a target bucket filled with compressed hay next to the hurdle barrier, directly in front of the stimuli array. Goats were encouraged to approach and investigate this bucket as from this position they would have a clear view of the photograph, unimpeded by the barrier.

During experiments, we affixed the appropriate photograph to a predetermined position at the front of the arena (top edge of image was 93.5 cm off the ground). This position ensured that the speaker (a Bose Soundlink Mini Bluetooth Speaker II), which was placed directly behind the image was at approximately mouth level (70 cm off the ground). The speaker used to broadcast playbacks in the current experiment has been confirmed to accurately reproduce the frequency range of the human voice (*Ben-Aderet et al., 2017*). We separated off the front of the arena using hurdles, imposing a distance between the subject and stimuli set of a minimum of 2.4 m. Goats have good visual acuity, being able to discriminate small details (3.4 cm symbols) at distances of 1.5–2 m (*Blakeman & Friend, 1986*).

## Habituation phase

We habituated subjects to staying in the experimental arena alone for extended periods over three, four-minute sessions that took place prior to testing, and occurred between a minimum of three hours and a maximum of eight days apart (mean ± SD = 2 days 6 hrs ± 1 day 21 hrs). The variation in interval between successive habituation sessions

was necessary as goats were not always willing to be led to the experimental enclosure or were displaced by more dominant individuals en route. Before starting each habituation session, we equipped goats with a Zephyr™ BioHarness 3.0, which in conjunction with AcqKnowledge v. 4.4.2 software (BIOPAC System Inc.) transmitted live cardiac data to a laptop (HP ProBook 650 G4) *via* Bluetooth during experimental trials. If subjects exhibited signs of aversion or aggression while this device was being fitted, we stopped and restricted future measurements taken during such trials to behaviour only (cardiac data collected from 15 out of 26 goats; for further details see Table S1).

For the first habituation session, an experimenter sat still with their head facing downwards at the front of the arena, where the photograph would be placed during experimental trials. As initially goats were not completely isolated, we aimed for this to reduce neophobia and encourage subjects to investigate their surroundings. For the following two sessions, subjects were kept alone in the arena. We occasionally repeated sessions when previous ones were halted due to adverse conditions such as rain, or when a long period of time had elapsed since the goat's third habituation session (approximately seven days). For all three habituation sessions, we ensured animals had access to water and small food rewards (dry pasta), the latter of which were placed in and around the target bucket to encourage them to investigate and spend more time at the front of the arena. The habituation phase preceded the test phase by a minimum of three hours and 15 min and a maximum of eight days (median = 1 day; interquartile range, IQR = 1 day 4 hrs 38 mins).

## Training phase

The training phase took place directly before each experimental trial and aimed to incentivise goats to approach and investigate the target bucket during the test phase. Goats entered the enclosure through the preparation pen where we fitted them with the BioHarness, which, subjects had become accustomed to wearing during habituation sessions. To attain a clearer ECG trace, we clipped a patch of fur around the left shoulder blade at least one day prior to testing, over which the BioHarness module would be positioned. ECG gel was applied liberally to sensors on the BioHarness belt before it was placed around the subject's thorax to improve conductance to the skin. Once a clear ECG trace had been secured, we led goats into the holding pen to begin training.

For training trial 1, Experimenter 1 and the subject were positioned inside the holding pen. Experimenter 2 held a piece of pasta (or cracker if goats were not sufficiently motivated by pasta) near the subject's head and once they had noticed it, walked slowly backwards towards the bucket while actively maintaining their attention. They then slowly and overtly placed two pieces of pasta inside, before crouching by the bucket and looking at it. Experimenter 1 then released the subject from the holding pen and stood aside, allowing the goat to pass and retrieve the pasta. The subject consumed both pieces of pasta before being led back into the holding pen. Training trial 2 proceeded similarly to the first, but instead of crouching down by the bucket, Experimenter 2 released the goat from the holding pen and stood aside to allow them to retrieve the reward. For training trials 3 and 4, Experimenter 2 gained the subject's attention from the bucket, before baiting it and releasing them from the holding pen. Goats passed a trial if they successfully reached the bucket within 30 s

after release and needed to pass all four trials consecutively to proceed to the test phase. If a subject failed to pass a trial, the training process was repeated from the beginning and if they were not sufficiently motivated to participate in training, they were released, and the experimental trial was carried out on a different occasion.

### Test phase

Goats experienced four experimental trials each (3–14 days apart; mean = 7 days). Two trials were congruent (face and voice were from the same person) and two incongruent (from different people), with subjects experiencing all combinations of photograph and voice playbacks from a preferred person and a caretaker in a random order (randomisation carried out using sample function in R: *R Core Team, 2020*).

Once training had been completed, we led subjects back into the preparation pen and following checks to ensure the ECG signal had been maintained, they were distracted while the photograph was positioned. The goat was then led back into the arena and the holding pen shut behind them. Experimenters 1 and 2 quietly hid behind the visual barrier in the preparation pen and monitored subject movement in a nearby video camera's LCD monitor (Fig. 1). We expected goats to approach and inspect the bucket (no pasta or water available during test phase) and when they were positioned close to and facing the front, we initiated the playback sequence (mean maximum amplitude ± SD = 76 dB ± 2 measured 1m away under field conditions using a CEM™ DT-8851 sound-level meter). Goats listened to a single sequence of two voice playbacks, with behavioural and cardiac responses being measured for 10 s following the initiation of each playback. If a subject moved away within the five seconds preceding the first playback, where possible the playback sequence was aborted, and reinitiated when they were in a good position. If goats failed to be in a suitable position for six minutes following trial initiation ($n = 11$ trials), the trial was suspended, and they were released. We placed an event marker in the ECG trace to indicate occurrence of each playback for later analysis of cardiac measures. Goats that had heard both playbacks were released 30 s after the final playback (median duration in arena = 1 min 20.95 s; IQR = 64.12 s). Of the 33 subjects tested, we excluded one from further testing due to health concerns, one for repeatedly failing to be in a suitable position for stimuli presentation, and a further subject and two trials from another due to a lack of training motivation.

### Video coding

A single observer coded behavioural data using BORIS v. 7.8.2 (*Friard & Gamba, 2016*), with goat latency to look towards the photograph and looking duration measured for 10 s following the initiation of each voice playback defined to the nearest frame (0.04 s). Goat looking behaviours were coded from when the goat's head was turned in the direction of the stimuli array, until they turned away. Behavioural measurements were extracted from footage captured from the front and back of the arena, before being compared and combined. Measurements taken from the front were often clearer and without a blind spot, so these took precedence when quantifying behaviour. However, this was not always possible, with technical issues in the front camera for three trials and the back camera for two preventing experiments from being successfully captured by both, so in these cases

we coded behaviours from the footage available. For one of these trials, the subject went into the camera's blind spot, so only behaviours that could be coded with certainty were recorded, with others defined as missing. In one further trial, the photograph fell down during testing, so it was also excluded from analysis.

Although we analysed behaviour coded by a single observer, a second observer, who was blind to the experimental condition, independently extracted behavioural measurements from 20% of trials. Agreement between observers proved to be high for goat duration of looking (concordance correlation coefficient: 0.917, 95% CI [0.849–0.956]) and latency to look (0.902, 95% CI [0.816–0.949]; DescTools package: *Signorell, 2017*). However, Bland-Altman plots indicated differences in latency measurements for observations where goats took longer to look.

### Exclusion criteria

Prior to analysis, we applied three exclusion criteria to determine which trials should be included in our final data set. Firstly, if goats failed to look during both response periods in the playback sequence, the trial was excluded as subjects were interpreted as not being sufficiently attentive to human cues to notice incongruencies between them ($n = 21$ trials: $n = 6$ congruent and $n = 15$ incongruent). Secondly, if goats were already looking towards the stimuli array before a playback had been initiated, these trials were excluded as goats were considered unlikely to be responding to differences in congruency between visual and vocal cues ($n = 23$ observations: $n = 16$ congruent and $n = 7$ incongruent; $n = 4$ trials: $n = 2$ congruent and $n = 2$ incongruent). Finally, once the first and second exclusion criteria had been applied, we also excluded goats that did not look for at least one congruent and one incongruent trial as these prohibited within-subjects comparison ($n = 3$ subjects excluded). Ultimately, we analysed 85 trials from 26 subjects ($n = 45$ congruent and $n = 40$ incongruent trials; see Table S1 for more information), eight experiencing stimuli from men, and 18 from women.

### Data analysis

#### General model parameters

Model simplification approaches, including information-theoretic ones (Akaike or Bayesian Information Criterion) can result in type 1 errors (*Forstmeier & Schielzeth, 2011*; *Forstmeier, Wagenmakers & Parker, 2017*). So instead, as in addition to congruency, there were several variables which could have had profound effects on goat behavioural and physiological responses we used a full model approach, as recommended by, for example, *Forstmeier & Schielzeth (2011)*.

Our primary variable of interest for all models was congruency, *i.e.,* whether the face and voice presented were from the same person (congruent), or from different people (incongruent). We also investigated whether goat responses changed over the playback sequence (playbacks 1 and 2), as could happen owing to, for example, habituation. In addition, we explored the interaction between congruency and playback number as there may be a lag before goats register and respond to incongruencies between stimuli presented. This interaction term was included in all models to begin with but removed if its effect was not significant to allow us to interpret the effects of playback number and congruency

separately (*Engqvist, 2005*). Further, as the experimental enclosure was erected at different locations in 2020 and 2021 and the pool of human stimuli collected was expanded in 2021, we controlled for the effect of year on goat responses. The goat's sex and gender of the person whose identity cues were presented were also considered (*Bognár, Iotchev & Kubinyi, 2018*; *Proops & McComb, 2012*; *McComb et al., 2014*; *Shih et al., 2020*), as was the identity of the photograph and voice, specifically whether or not they were derived from the preferred person. Goats experiencing an extended time in the experimental enclosure may have had longer to habituate or become more aroused due to prolonged isolation from conspecifics (*e.g.*, *Siebert et al., 2011*; *Briefer, Tettamanti & McElligott, 2015*), so the interval between the experimenters leaving the arena (trial beginning) and the onset of the playback sequence (preliminary duration) was included as a covariate. Finally, noise in the ECG trace meant heart rate and heart rate variability (HRV) often could not be calculated over the entire response period. As the time in which it was possible to calculate cardiac responses varied with ECG signal quality, measurement period potentially represented an important control variable, so was included as a covariate in all relevant models (*e.g.*, *Reefmann, Wechsler & Gygax, 2009*; *Briefer, Oxley & McElligott, 2015*). As for random effects, trial number (1–4) was nested within subject identity to control for repeated measurements taken from each goat both within and between trials, as was the unique identifier given to the photograph and playback sequence used for an experimental trial (the same stimuli were used over multiple trials).

To summarise, the following variables were used to model goat latency to look and looking duration following each playback, as well as heart rate and HRV:

$$\text{Goat Response}_{(\text{Behaviour/Cardiac})} = \text{Congruency}_{(\text{Congruent/Incongruent})} + \text{Playback No.}_{(1/2)} +$$
$$[\text{Congruency x Playback No.}] + \text{Year}_{(2020/2021)} + \text{Goat Sex}_{(\text{Female/Male})} + \text{Human}$$
$$\text{Gender}_{(\text{Female/Male})} + \text{Photograph ID}_{(\text{Preferred/Non-preferred Person})} + \text{Preliminary Duration}$$
$$+ [\text{Measurement Period}] + (1|\text{Subject/Trial No.}) + (1|\text{Face ID}) + (1|\text{Voice ID}).$$

Variables enclosed in square brackets were included only in models predicting cardiac responses. All analyses were conducted using R v. 4.3.3. (*R Core Team, 2024*).

### Latency to look following playbacks

We removed observations in which goats did not look following a playback from our latency analysis to avoid ceiling effects from assigning instances where goats did not look the maximum latency (10 s; $n = 29$ observations: $n = 15$ congruent and $n = 14$ incongruent; similar approach used in, *e.g.*, *Pitcher et al., 2017*). In keeping with the third exclusion criteria, two additional goats were removed from our latency analysis. Using the R package glmmTMB (*Brooks et al., 2017*), we fitted models assuming different error structures (*e.g.*, linear, binomial and gamma), assessing the suitability of each using the DHARMa package (*Hartig, 2021*) and identifying the best fit model using Akaike's Information Criterion (AIC). Ultimately, we analysed latency using a lognormal mixed model (log link), removing the covariate, preliminary duration and random effect, photograph used to resolve issues with model convergence. Our final sample size when analysing goat latency to look comprised 24 goats taking part in 78 trials ($n = 42$ congruent and $n = 36$ incongruent; $n = 119$ observations).

### Looking duration following playbacks

We compared the fit of multiple models assuming different error structures to goat looking duration responses using the package glmmTMB (*e.g.*, linear, binomial, zero-inflated negative binomial and gamma models: *Brooks et al., 2017*), assessing the suitability of each model using the DHARMa package (*Hartig, 2021*). According to AIC we found that the best fit model had a Tweedie distribution (*Tweedie, 1984*; *Bonat & Kokonendji, 2017*), that assumes positive values and enabled us to account for zero-inflation in our duration responses due to the high number of goats that did not look up following voice playbacks. The covariate, preliminary duration and random effect, photograph were removed due to issues with model convergence. Our final sample size for the duration analysis consisted of 26 goats participating in 84 trials ($n = 45$ congruent and $n = 39$ incongruent; $n = 154$ observations).

### Heart rate & heart rate variability

We calculated heart rate as beats per minute (BPM) and HRV as the root mean square of successive differences between heartbeats multiplied by 1000 (RMSSD). Changes in both these measures are thought to indicate shifts in physiological arousal and have been shown in relation to a variety of contexts in goats (*e.g.*, *Briefer, Tettamanti & McElligott, 2015*; *Briefer, Oxley & McElligott, 2015*; *Baciadonna, Nawroth & McElligott, 2016*; *Baciadonna, Briefer & McElligott, 2020*). Baseline heart rate and HRV was calculated ideally in the 10 s preceding the onset of the playback sequence. Trials began when both experimenters had left the arena and to reduce the effect of human manipulation on cardiac responses, baseline heart rate and HRV were calculated only following this, which for some observations ($n = 13$ trials) meant the measurement period was less than 10 s. When noise present in the ECG trace restricted this period to less than five seconds, where possible (preliminary duration before onset of a playback sequence was greater than 10 s), we expanded the time frame for calculating this baseline up to 30 s prior to the first playback until a measurement period of 10 s could be achieved ($n = 7$ trials).

We examined the difference in heart rate and HRV calculated in the response periods following playbacks 1 and 2 compared to the baseline period ($\Delta HR = HR_{pb1} - HR_{baseline}$ or $\Delta HR = HR_{pb2} - HR_{baseline}$ | $\Delta HRV = HRV_{pb1} - HRV_{baseline}$ or $HRV_{pb2} - HRV_{baseline}$). These measurements were used as variation in baseline heart rate and HRV meant relative changes in these responses measured over a single trial were more meaningful than absolute differences between individuals and trials (mean baseline BPM $\pm$ SD = 115.15 $\pm$ 16.97; mean baseline RMSSD $\pm$ SD = 22.42 $\pm$ 25.57). Model residuals conformed to an approximately normal error structure (assessed using DHARMa package: *Hartig, 2021*), so we fitted a linear mixed model (LMM) to goat heart rate responses (lmer function, lme4 package: *Bates et al., 2015*). However, having fit a LMM to HRV data and visualised its residual variance, plots indicated the presence of extreme values.

Using a z-score method (outliers package), we identified HRV values falling outside the 95% quantiles. After excluding these observations ($n = 8$), LMM assumptions were met so we used this approach to model shifts in HRV relative to congruency accordingly (lme4 package: *Bates et al., 2015*). Post hoc tests were conducted using the emmeans package with

**Table 1  Predictors of time taken for goats to look at the facial photograph following playbacks of a familiar person's voice (lognormal GLMM).** Results concerning the primary effect of interest, congruency are shown in bold.

| Explanatory variable | β | S.E. | z-value | p-value |
|---|---|---|---|---|
| Intercept | 0.534 | 0.185 | | |
| **Congruency (I)[a]** | **−0.005** | **0.097** | **−0.051** | **0.960** |
| Playback No. (2)[b] | −0.051 | 0.098 | −0.522 | 0.602 |
| Year (2021)[c] | 0.001 | 0.107 | 0.007 | 0.995 |
| Goat Sex (M)[d] | 0.120 | 0.135 | 0.887 | 0.375 |
| Human Gender (M)[d] | 0.088 | 0.121 | 0.725 | 0.468 |
| Photograph ID (P)[e] | 0.034 | 0.097 | −0.348 | 0.728 |
| Voice ID (P)[e] | −0.150 | 0.102 | −1.475 | 0.140 |

Notes.

Key: I = Incongruent; M = Male; P = Preferred Person. Reference Categories: a = Congruent; b = Playback Number 1; c = 2020; d = Female; e = Non-preferred Person.

Tukey's corrections to account for multiple comparisons (*Lenth, 2021*). Our final sample size for heart rate analysis comprised 15 goats participating in 45 trials ($n = 23$ congruent and $n = 22$ incongruent trials: $n = 81$ observations) and in 44 trials for the HRV analysis ($n = 74$ observations; $n = 22$ congruent and 22 incongruent).

# RESULTS

## Latency to look following playbacks
The time taken for goats to look towards the photograph following the presentation of a voice playback was not significantly affected by congruency between these human cues (Table 1; Fig. 2A). Playback number (1 or 2), the year a goat was tested (2020 or 2021), their sex, the gender of the human stimuli presented, and whether the voice or the photograph came from a preferred person also did not affect their latency to look.

## Looking duration following playbacks
Congruency between the human face and voice presented did not significantly affect how long goats looked towards the stimuli array following presentation of voice playbacks. Neither did playback number, the year a goat was tested, their sex, the gender of the human cues presented or whether the face and voice shown belonged to a preferred person (Table 2; Fig. 2B).

## Heart rate & heart rate variability
Changes in goat heart rate (Fig. 3A) and HRV from baseline values (Fig. 3B) were observed over the playback sequence (congruency x playback number interaction significant: Table 3), with heart rate decreasing between playbacks 1 and 2 when the human face and voice presented were incongruent ($β ± s.e. = 7.198 ± 1.641$, t-ratio = 4.385, $p = 0.0006$: Table S5). The results were broadly similar between the full models presented (Table 3) and more basic models containing only the effect of the congruency and playback number interaction, and the random effect of trial number nested within the goat being tested (for results of basic models, see Tables S2–S4). No significant changes in heart rate were

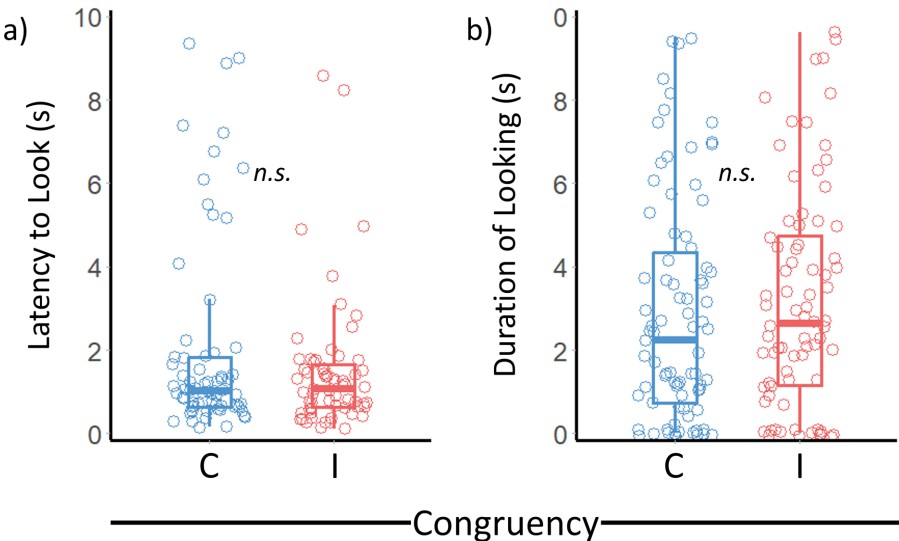

**Figure 2 Effect of congruency between human facial and vocal cues on goat looking behaviours.** Median and interquartile range (IQR) for the effect of congruency on (A) how long it took for goats to look and (B) how long they looked at the photograph following each voice playback. Boxplot whiskers extend to the maximum and minimum value less than 1.5 times above or below the IQR respectively. C = Congruent; I = Incongruent. *n.s.* = non-significant.

**Table 2 Predictors of how long goats looked at the photograph following voice playbacks (Tweedie GLMM).** Results concerning the primary effect of interest, congruency are shown in bold.

| Explanatory variable | β | S.E. | z-value | p-value |
|---|---|---|---|---|
| Intercept | 1.142 | 0.245 | | |
| **Congruency (I)[a]** | **0.055** | **0.142** | **0.388** | **0.698** |
| Playback No. (2)[b] | −0.191 | 0.135 | −1.412 | 0.158 |
| Year (2021)[c] | −0.101 | 0.198 | −0.510 | 0.610 |
| Goat Sex (M)[d] | 0.205 | 0.225 | 0.915 | 0.360 |
| Human Gender (M)[d] | −0.044 | 0.210 | −0.211 | 0.833 |
| Photograph ID (P)[e] | 0.029 | 0.142 | 0.205 | 0.837 |
| Voice ID (P)[e] | −0.202 | 0.160 | −1.274 | 0.203 |

**Notes.**

Key: I = Incongruent; M = Male; P = Preferred Person. Reference Categories: a = Congruent; b = Playback Number 1; c = 2020; d = Female; e = Non-preferred Person.

observed when goats were shown congruent cues, nor were there any significant pairwise changes in HRV between playbacks 1 and 2 based on congruency (Tables S5; S6). However, when there was a longer measurement period over which to calculate goat HRV, we found that these values were significantly lower than when shorter periods were available (Table 3).

We applied the same exclusion criteria for our cardiac data set as for our behavioural one. Specifically, trials were excluded when goats did not look for both response periods of the playback sequence, and goats when they did not look for one congruent and one incongruent trial. Additionally, trials were excluded when goats were looking before

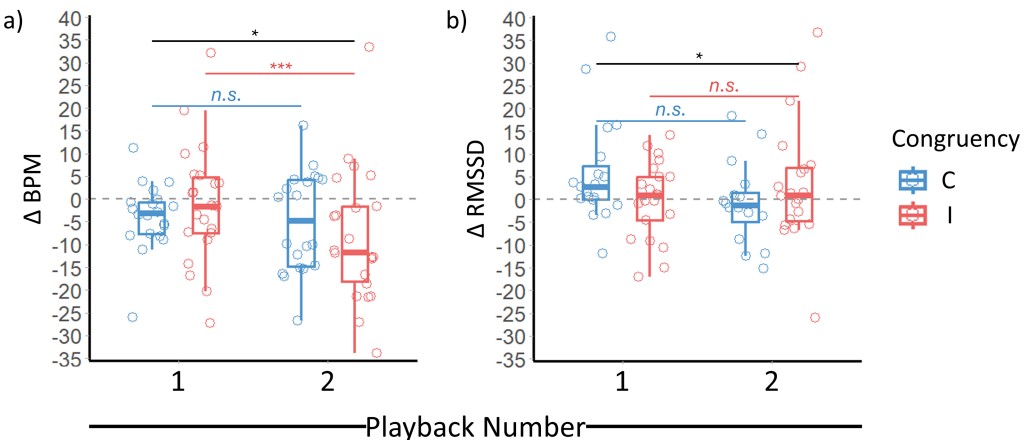

**Figure 3** **Effect of congruency between human visual and vocal identity cues on goat cardiac responses.** (A) Shifts in goat heart rate and (B) HRV from baseline values (measured before onset of playbacks) as a function of face-voice congruency and playback number. Boxplots feature median and IQR. Whiskers extend to the maximum and minimum value less than 1.5 times above or below the IQR respectively. Baseline values are indicated by the dashed grey line. BPM = Beats per minute; C = Congruent; I = Incongruent. Significance for the interaction between congruency and playback number shown at the top in black, with that of further pairwise interactions being shown underneath. $*p < 0.05$; $**p < 0.01$; $***p < 0.001$ n.s. = non-significant.

**Table 3** **Predictors of goat heart rate and HRV, relative to baseline values (measured before onset of a playback sequence) in the response periods following presentation of a familiar human voice (LMM).** Results concerning the primary effect of interest, the congruency x playback number interaction are shown in bold.

| Explanatory variable | Heart rate (BPM) | | | | Heart rate variability (RMSSD) | | | |
|---|---|---|---|---|---|---|---|---|
| | $\beta$ | S.E. | $t$-value | $p$-value | $\beta$ | S.E. | $t$-value | $p$-value |
| Intercept | −9.133 | 10.361 | | | 40.435 | 12.036 | | |
| Congruency (I)[a] | 1.253 | 3.700 | 0.339 | 0.737 | −6.421 | 3.488 | −1.841 | 0.071 |
| Playback No. (2)[b] | −0.756 | 1.822 | −0.415 | 0.681 | −6.353 | 3.494 | −1.818 | 0.074 |
| **Congruency (I)[a] x Playback No. (2)[b]** | **−6.442** | **2.455** | **−2.624** | **0.013**[*] | **10.605** | **4.711** | **2.251** | **0.028**[*] |
| Year (2021)[c] | 5.685 | 3.899 | 1.458 | 0.177 | −5.089 | 4.025 | −1.264 | 0.234 |
| Goat Sex (M)[d] | −4.661 | 4.185 | −1.114 | 0.288 | 0.466 | 4.179 | 0.112 | 0.913 |
| Human Gender (M)[d] | 1.411 | 4.163 | 0.339 | 0.741 | −0.097 | 4.213 | −0.023 | 0.982 |
| Photograph ID (P)[e] | 5.728 | 3.409 | 1.681 | 0.102 | −0.630 | 2.489 | −0.253 | 0.801 |
| Voice ID (P)[e] | −4.152 | 3.459 | −1.200 | 0.239 | 0.228 | 2.473 | 0.092 | 0.927 |
| Preliminary Duration | 0.015 | 0.024 | 0.642 | 0.525 | −0.008 | 0.020 | −0.416 | 0.679 |
| Measurement Period | 0.321 | 0.991 | 0.324 | 0.747 | −3.491 | 1.232 | −2.832 | 0.006[**] |

**Notes.**
Key: I = Incongruent; M = Male; P = Preferred Person. Reference Categories: a = Congruent; b = Playback Number 1; c = 2020; d = Female; e = Non-Preferred Person.
[*]$p < 0.05$.
[**]$p < 0.01$.

a playback had been initiated. We considered this latter criterion to more likely effect behavioural over cardiac measures. Therefore, to evaluate the robustness of the effect congruency and playback number on goat cardiac responses, we added observations where goats had already been looking before the onset of a playback back into our cardiac data

set. However, when these observations were reinstated, congruency, in combination with playback number no longer affected goat heart rate ($\beta \pm$ s.e. $= -3.426 \pm 2.214$, t $= -1.547$, $p = 0.129$) or HRV ($\beta \pm$ s.e. $= 5.220 \pm 4.333$, $t = 1.205$, $p = 0.233$).

## DISCUSSION

We presented goats with a photograph of a familiar person and a voice that either matched (was congruent with) or did not match (was incongruent with) that photograph's subject. If they could recognise the familiar person cross-modally, goats were predicted to respond faster and for longer when there were incongruencies between stimuli, reflecting a violation of their expectations (*e.g.*, *Adachi, Kuwahata & Fujita, 2007*; *Takagi et al., 2019*). Contrary to what was predicted, whether the photograph and voice were taken from the same or different people had no effect on how fast or long goats looked towards the photograph following each playback. That being said, congruency in combination with playback number (1 or 2) did affect cardiac responses (heart rate and HRV), with a decrease in heart rate being observed as the playback sequence progressed only when goats were experiencing incongruent cues. However, we found evidence to suggest changes in goat cardiac parameters with congruency may not have been robust and there were no changes observed in our primary variables of interest, behaviour (*e.g.*, *Adachi, Kuwahata & Fujita, 2007*; *Takagi et al., 2019*). Although our findings could suggest that goats had successfully perceived differences in congruency between the visual and vocal information presented, further evidence is needed to determine whether they are capable of cross-modal recognition of humans.

When the face and voice presented came from different people, goat heart rate decreased between playbacks 1 and 2, which may reflect a lag before changes in cardiac response could be enacted. The decrease in heart rate observed in the incongruent, but not congruent condition could reflect the heightened attention to external stimuli elicited by such novel face-voice combinations. When the heart beats, baroreceptors in the walls of the aortic and carotid arteries fire, with these signals then being transmitted to the brain *via* the glossopharyngeal and vagus nerves (*Skora, Livermore & Roelofs, 2022*). Although integral for modulating heart rate, this process also represents a potential source of noise acting to attenuate brain activity. Parasympathetically-mediated decelerations in heart rate reduce the frequency of baroreceptor signals, potentially enabling goats to gather and process more information from their environment following these unexpected pairings of human identity cues. In addition to heart rate, we observed a change in HRV with congruency, although we were unable to disentangle precisely where differences lay. However, changes in cardiac responses were only observed when applying the same exclusion criteria used for analysing goat behaviour. When we reinstated observations where goats were already looking before the start of playbacks, the effect of congruency on goat cardiac responses disappeared, which could suggest the changes observed were not robust.

One possible explanation for the lack of robust differences in cardiac response with congruency could be our limited sample size. Some goats expressed signs of aversion or aggression while the heart rate monitor was being fitted, with us ultimately only being

able to collect cardiac responses in 15 of our 26 goats. Observations used in analysis of HRV were further reduced by outliers in our data set (HRV was highly variable between goats and sessions). In addition, the measurement period over which cardiac responses were calculated was relatively short (less than 10 s) and variable due to noise within the ECG trace, potentially limiting accuracy in which these parameters were measured. In humans, accurate measurements of HRV may be achievable within 30 s, although five minutes is more conventionally advised (*Salahuddin et al., 2007*; *Shaffer & Ginsberg, 2017*), with the latter also recommended in horses (*Stucke, Ruse & Lebelt, 2015*). Accordingly, the length of measurement period affected goat HRV, suggesting this factor influenced the accuracy in which HRV was calculated. Heart rate, although a simpler measure than HRV, would also likely have been measured more accurately given a longer sampling period. Finally, although investigators did not measure cardiac parameters, similar congruency paradigms to the one used here have found a range of species tend to look quicker and/or for longer when presented with incongruent conspecific cues (*e.g.*, *Proops, McComb & Reby, 2009*; *Gilfillan et al., 2016*; *Baciadonna et al., 2021*) and, moreover, human identity cues in different modalities (*e.g.*, *Adachi, Kuwahata & Fujita, 2007*; *Takagi et al., 2019*; *Lampe & Andre, 2012*).

We believe goat behaviour may not have changed as expected, along with the lack of robust changes in physiology could be either due to factors related to goat social cognition or to our experimental design. Firstly, in order to register incongruencies, animals need to have developed an internal template for known individuals (*Proops, McComb & Reby, 2009*; *Ratcliffe, Taylor & Reby, 2016*). Not all goats in our investigation may have possessed such a template for both people they were experiencing cues from, either through lack of cognitive ability or familiarity with their individual-specific cues. Indeed, assuming goats are more likely to develop complex cue use to recognise humans they have a strong (positive) relationship with (*Pitcher et al., 2017*), future researchers may consider using formal preference tests to, for example, assess animal readiness to interact with different familiar people (*e.g.*, human approach tests; *De Passillé et al., 1996*; *Sankey et al., 2010*; review: *Rault et al., 2020*). Secondly, we used photographs instead of live people. Photographs exclude olfactory, body (facial photographs were used), depth, perspective, and motion cues and alter colour, all of which limits the amount of information goats would have had available to distinguish between individuals (*Hill, Schyns & Akamatsu, 1997*; *Poindron et al., 2007*; *Fagot & Parron, 2010*; *Keil et al., 2012*; *Lansade et al., 2020*). Aside from it being more difficult for non-human animals to recognise objects from photographs, in order to have registered incongruencies between the visual and vocal information presented, goats would have also had to treat the photographs as representations of the people they depict (*Fagot & Parron, 2010*). A recent investigation found that goats did not express an immediate preference for photographs of group members over unknown conspecifics, nor did they learn at a faster rate when required to select a group member from three unknown individuals than vice versa (*Langbein, Moreno-Zambrano & Siebert, 2023*). These findings were interpreted as goats being unable to spontaneously link these photographs to their real-life counterpart, although it was suggested that they learnt to do this following presentation of different photographs of the same individual. Furthermore, the static,

unresponsive nature of images can mean they are less salient and more rapidly habituated to than live stimuli (*Vandenheede & Bouissou, 1994*; *Bovet & Vauclair, 2000*). Indeed, the prolonged presentation of photographs, rather than displaying visual and vocal cues in two short stages (similar to: *Adachi et al., 2006*; *Adachi, Kuwahata & Fujita, 2007*; *Adachi et al., 2009*; *Takagi et al., 2019*) may have provided the opportunity for goats to inspect the 2D facial image more thoroughly. Realising the image was static, they then may have been less able to pair it with the dynamic vocal information presented (based on comments made by an anonymous reviewer). Similar investigations in goats and other species may consider using live people, or even videos as stimuli to discriminate between (*e.g.*, *Proops & McComb, 2012*; *Trösch et al., 2020*). Given the significance of humans in the lives of goats and other domesticated animals further research is needed to understand how and when they discriminate among humans.

Robust recognition abilities like cross-modal recognition, enabling animals to discriminate among humans, as well as conspecifics, may allow better tracking of interactions with certain people, thereby forming a basis for interspecific social relationships (*Tibbetts & Dale, 2007*; *Hemsworth & Coleman, 2011*; *Wiley, 2013*; *Yorzinski, 2017*). Negative human-animal relationships, resulting from a negative perception of humans, have been linked to poor welfare, with fear of humans primarily being the driving factor (reviewed by: *Mota-Rojas et al., 2020*). In contrast, in a positive human-animal relationship, social interactions with certain people may develop rewarding properties, providing opportunities for animals to experience positive emotions, buffer against stressful situations (*e.g.*, husbandry procedures) and potentially increase an animal's long term stress resilience (reviewed by: *Rault et al., 2020*). How well goats can discriminate among humans will affect whether experiences with certain people are attributed to that individual (individual recognition), people sharing similar features (class-level recognition, *e.g.*, vets *versus* regular caretakers) or even just to humans in general (although recognition and generalisation are not mutually exclusive: *Brajon et al., 2015*; *Yorzinski, 2017*). Indeed, if goats had registered incongruencies between human stimuli presented in different modalities it would suggest the presence of an internal representation for familiar people, a prerequisite for individual recognition (*Proops, McComb & Reby, 2009*). Further research is needed to better understand the specificity and mechanisms that goats use to discriminate among people.

In conclusion, the differences in response towards congruent and incongruent human identity cues could suggest that goats were responding to changes in stimuli congruency at least at physiological levels. However, further support is needed to establish whether they can cross-modally recognise humans. Future investigations may consider use of more dynamic human representations to act as cues, for example, live humans, or video footage. Establishing whether goats are capable of cross-modal recognition would be important not only for furthering our basic knowledge of social cognition in human-animal relationships, but could also have vital applied implications for better understanding and ultimately improving the welfare of domesticated animals.

## ACKNOWLEDGEMENTS

We are very grateful to all of our field assistants and especially to Karen Lorena Estupinan Cely, Anastasia Grimes, Ellen Ye and Daniella Bernal Vega. We would also like to thank Dr Maarit Mäenpää, Dr Gavin Simpson and Jia Liu, our reviewers, Dr Holly Root-Gutteridge, Dr Leanne Proops and the anonymous reviewer and editor, Dr Jennifer Vonk for their comments and advice. We are further thankful to everyone who provided photographs and voice samples for our experiments, and to the staff and volunteers at Buttercups Sanctuary for Goats for their help and advice and allowing us access to their goats.

### Funding

This research was supported by the Kimmela Centre for Animal Advocacy, Ede & Ravenscroft and the University of Roehampton. The funders had no role in study design, data collection and analysis, decision to publish, or preparation of the manuscript.

### Grant Disclosures

The following grant information was disclosed by the authors:
Kimmela Centre for Animal Advocacy, Ede & Ravenscroft and the University of Roehampton.

### Competing Interests

Alan G. McElligott is an Academic Editor for PeerJ.

### Author Contributions

- Marianne A. Mason conceived and designed the experiments, performed the experiments, analyzed the data, prepared figures and/or tables, authored or reviewed drafts of the article, and approved the final draft.
- Stuart Semple conceived and designed the experiments, authored or reviewed drafts of the article, provided supervision, and approved the final draft.
- Harry H. Marshall analyzed the data, authored or reviewed drafts of the article, provided supervision, and approved the final draft.
- Alan G. McElligott conceived and designed the experiments, authored or reviewed drafts of the article, provided supervision, and approved the final draft.

### Human Ethics

The following information was supplied relating to ethical approvals (i.e., approving body and any reference numbers):

The University of Roehampton Life Sciences Ethics Committee gave approval to collect photographs and voice samples from human volunteers (Ref. LSC 19/ 280).

### Animal Ethics

The following information was supplied relating to ethical approvals (i.e., approving body and any reference numbers):

Our stimuli collection from human participants and all experimental procedures were approved by the University of Roehampton Life Sciences Ethics Committee (Ref. LSC 19/280), with the latter being in line with ASAB guidelines for the use of animals in research (*ASAB/ABS, 2020*).

## Data Availability

The raw data of goat behavioural and cardiac responses and scripts are available in the Supplemental Files.

## Supplemental Information

Supplemental information for this article can be found online at http://dx.doi.org/10.7717/peerj.18786#supplemental-information.

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
