# Peer review of "Do goats recognise humans cross-modally?"

_PeerJ, doi:10.7717/peerj.18786_

## Round 0.1 · original submission · Major Revisions

I was fortunate to receive reviews of your work from three experts, all of whom appreciated the value of your work and found the manuscript to be well-written. However, each reviewer pointed to the need for some clarification, particularly regarding your recording of latency and duration measures. It may be necessary for you to recode and re-analyze your data in light of these comments. We all appreciated that you were tentative in your conclusions. I also appreciated the detail in the statistical analysis section.

The reviewers note that photographs may not have been the ideal way to present visual cues. I have an even more serious concern with the use of photographs. I realize that photographs have been used in similar paradigms in the past, but it always strikes me as a bit strange. Why would an animal expect to hear the voice of a person that is presented in a photograph? Are they supposed to anticipate that the photograph is the actual person and therefore may speak? In the discussion, you indicate that goats may not see photographs as representative of real-life objects, which is a major limitation. Even if they had this ability, should they not be able to discriminate between photographs and real people? Especially in the absence of olfactory cues? I think the strangeness of the situation should be carefully addressed - at least in the discussion. In light of this issue, your data really cannot speak to whether goats recognize individuals at all. Instead, your results seem to corroborate that photographic stimuli may not be appropriate for experiments with goats. Your study would be strengthened if you had presented olfactory cues along with auditory cues. Even if you had obtained the expected effects, differences in looking on incongruent trials could be due to the fact that cues are coming from two people rather than one person, rather than the goat having an integrated representation of one individual. You should cut down some of the discussion of the value of individual recognition given the fact that your data cannot speak to this ability. Obviously you cannot change the paradigm that you chose, but I think you should be even more circumspect with its limitations in your discussion and focus more on future directions than on conclusions regrading recognition.

I am also concerned with the fact that goats were presented with only two human stimuli (if I read this correctly). Why did you introduce the confound of preferred versus non-preferred people rather than choosing two individuals that were equally preferred and familiar for pairing the stimuli? Any differences in looking may have been due to preferences for the voice or the image of a preferred person. Even though you did not find evidence of such an effect, you created the possibility that goats could be responding to preference rather than recognition.

Throughout, please watch the placement of “only.” For example, in the abstract (line 42), “differences were only observed” should be “differences were observed only in…”

Please use commas before “which” (e.g., line 110 and check throughout). On line 147, add a comma or change “which” to “that.” Line 152, 529 and 335, change to “that.”

I found lines 41-42 awkwardly written. Please rephrase.

I don’t think you need the phrase “cross-modally” on line 109. The point is that determining when cues from two or more modalities match or fail to match indicates a representation of the individual. Thus, cross-modal presentations are more of a tool to investigate recognition rather than the outcome.

I think that more of a rationale is needed for why you expected physiological arousal in the incongruent conditions.

·

Basic reporting

This paper presents interesting data on the ability of domestic goats to cross-modally recognise familiar humans. I’ve already had the pleasure of reviewing this paper once and am pleased to see the authors have included my suggestions in the revised document. Overall, It is well-written and clear in its objectives and outcomes.

The introduction is generally clear and well-thought out. I would like to ask that you add a little on the choice of still photograph vs live person or video as this ends up being a critical component of your findings compared to those for other species. As it’s an established method, if you could just add some studies that also used it, that would help the reader understand your choice.

Some minor points in the generally very good introduction:
Line 72, the information about why using more than one modality here seems odd in terms of how often are the goats interacting with people in darkness? Out of sight or the view blocked by a bush etc. seems like a more relevant example. This is a very small point though and one I’m happy to be persuaded from.
Similarly, while I understand your point that these are not working animals akin to dogs, any animal living under close supervision by humans, as goats do, surely gains an advantage from being able to recognise familiar individuals from a host of cues, so this seems to over-state the surprise of the selection pressure exerted here.
Could you also please make it clearer why the expectation was incongruency leads to higher arousal? Was this because you expected the goats to be confused as opposed to excited? I’m unclear.

Experimental design

The experimental design is solid. It's a shame that photos were chosen over videos due to what we know of goats' responses to photos as of 2023, but that information was not available when the experiments were completed.

Validity of the findings

While most of this is well-written, there are some concerns with how the behavioural measurements were made, which may be a case of confusing language rather than an actual issue.
My main concern is that the way it is written, my understanding is that the behaviours were only measured for animals from the end of playback and start of “10s response period” even though the response would have begun potentially as soon as the animals heard the voice in the trial, making a ~11.28s response period. Furthermore, the latency is calculated for animals who were already looking at the photographs when the response period started but that this occurred after the onset of playback. The playback itself was ~1.28s but this was not included in the “response period” so the goats may have been looking for 0 to 1.28s prior to the period, or could have glanced then looked away (unlikely given the time frame but possible). Have I understood this correctly?
I’d argue it’s not possible to include latency in this way, as the critical period is voice onset not after it is finished as the animal may not have been looking before voice onset, then moved as the voice started and quickly turned. It would be better to recode from the start of voice onset.
For latency, this has an issue of whether they have begun to move before the end of the playback. If so, how many of the animals was this true for? Were any looking before playback began? This seems an odd way to measure a response latency.
Similarly, duration is affected as differences in latency to look will affect how long they look for overall, though this is less affected by the metric.
I’m also curious as to what would happen if you analysed only the “positive gaze” duration responses. This looks at the duration of response when, and only when, the animals looked at the target and excludes the zeros from the responses. The model would then be limited to the responses where the goats did respond but would allow direct comparisons between how long they responded for.
Minor points
In the paragraph starting line 272, do you mean randomised or was this pseudo-random with counterbalancing so that you had relatively equal numbers from each group at each presentation position (1-4)? If not, how did you counteract the possible clumping of trials?
305 – I’m not clear on how many trials were lost here. Was it just one?
That’s very good inter-observer reliability, well done!
326 – I’m finding the way this is laid out a little confusing. Could you please add some kind of summary in Supplementary or elsewhere how many goats had 2, 3 or 4 trials included in the final data?
While I agree that full models are preferable to selective methods for finding “best” models, what happens if you compare your extensive model to a very simple one of Congruency + random effects? E.g., discarding everything that is not the actual experimental goal? This could be interesting to including in SI.

383 on – So are the durations flattened into 1s chunks or into a binomial model? The models you discuss do not usually cope well with non-integer numbers. This seems to imply you reduced duration to 0/1 rather than the actual time they looked. This is fine but I’m just not clear.
Tables 1 & 3 – if a p-value is less than <0.001, please just state it as such as it’s not possible to be as precise as <2x10-16 even with R packages. If you would like to explore a secondary method of underlining the strength of the result, S values are a better option – Rafi & Greenland (2020) offer an introduction to their use.
431 – Surely there’s a mistake here somewhere, an increase of “0.12%” is unlikely to have been significant. Did you mean seconds here? It appears it may be the case from your data. Ditto for 1.07% in 434.
439 – Please add stats here.

Additional comments

The discussion is well-written and carefully balances the single significant result with the caveats attached. I applaud the authors for how they have approached the challenge of this.

The paper is overall interesting, well written, clear and with an interesting result.

Cite this review as

Reviewer 2 ·

Basic reporting

The paper is well written and includes almost all the details necessary to understand the experimental design and the practicalities around the study. I particularly appreciate all the details around the ethics as well as the provision of data and code which was fully reproducible and as a result allowed the verification of the findings. Some notes that could improve the clarity of the write up:
a) Line 144-145: The authors state, “We selected a target sample size based on a power analysis of our 2020 results”. Please specify which results you are referring to, and if these are published results, please cite them.

b) Lines 169-172: “Goats tested in 2020 were subject to photographic and vocal stimuli from pairs of caretakers, but in 2021, one of the two people goats experienced stimuli from (the preferred person) was not responsible for carrying out husbandry procedures (e.g., hoof trimming), which goats likely perceived as aversive. […] This could have affected the nuance of the relationship with their preferred person for goats tested in 2020 compared to those tested in 2021.” Please clarify whether this affected your experimental design or interpretation of the results in any way.

c) Lines 183-184: “For vocal recordings, we asked each person to say the phrase: “Hey, look over here,” several times in a manner they would normally use to address goats.” Please explain a bit more about these stimuli, e.g., whether the voice was intonated or flat, how long each vocal sample was etc.

d) Lines 234-235: “For the first habituation session, an experimenter sat still with their head down at the front of the arena where the photograph would be placed during experimental trials [...]” Could you please explain why the experimenter was there to start with? Could it have an effect of drawing the goats’ attention to this area during testing?

e) Please be consistent when using terms such as playbacks vs. playback sequence vs. voice sample. If I understand correctly, one playback sequence consisted of 5s silence, vocal sample (“Hey, look over here”) 10s of silence, the repeat of the same vocal sample, 30s of silence. When in line 285 you say “a series of two playbacks” do you mean one playback sequence (with two identical voice samples) or two playback sequences (with different vocal sample in each)? Please try to label things consistently to avoid confusion.

f) Could you clarify what you mean by ‘looking behaviour’ in goats? How did you code it? (position of the head or of the eyes?)

g) Could you state clearly if coding of behaviour was carried out only after the first voice sample or after both? In lines 193-194 you refer to the 10s after the first voice sample as the “response period” however it appears that you also code behaviour after the second voice sample within the same playback sequence. If this is the case, it would be worth stating it explicitly (e.g., by referring to both periods of silence as “response periods”)

Experimental design

I think that the experiment is well designed and should in principle allow to answer the question that the authors are asking. However, the way videos were coded needs clarification.

A few comments below:

If I understand correctly, the behaviour was only coded within the 10s “response period” which starts after the first vocal sample (C1 or I1) and finishes when the repeat of the vocal sample is played (C2 or I2). Please explain why latency was not coded from the start of the vocal sample.

Presumably the goat could have looked at the picture within the duration of “Hey, look over here” and looked away (and not looked back before 10s response period started and finished) – you will have then coded it as 10s (didn’t look throughout).

Equally, if the goat was looking in the general direction of the picture anyway before the playback was initiated and not changed what it was doing when the playback started and finished, you will have coded it as ‘looked at 0s’ (even though its looking response could have had nothing to do with the vocal sample being played, it could just be positioned that way).

I would therefore recommend further clarification of why this way of coding was undertaken. The consequence of this coding also meant that you ended up with two peaks in the data.

I looked at your code to see how you made your latency variable binomial but I’m afraid I’m not familiar with the technique that you used. Could you perhaps explain it in more detail, so the reader understands what was done (as my initial thought was that you ended up aggregating values around 0s and 10s and this is not what you did).

Validity of the findings

I think the findings around the duration and the heart rate look fine, but the latency findings need further clarification, especially around the choice of the coding window. There may be a very good reason why it was done that way but the authors will need to explain it further so the reader can understand the reasoning behind this choice.

Additional comments

It might be worth discussing in a bit more detail the limitations of using the looking behaviour as a measure. There is a good recent paper by Wilson et al (Wilson, V. A. D., Bethell, E. J., & Nawroth, C. (2023). The use of gaze to study cognition: limitations, solutions, and applications to animal welfare. Frontiers in Psychology, 14(May), 1–10. https://doi.org/10.3389/fpsyg.2023.1147278) that goes into more detail and is perhaps worth citing.

Cite this review as

·

Basic reporting

This study is the first to assess whether domestic goats are capable of cross-modal individual recognition of familiar humans, and as such makes an important contribution to the . The background and references are appropriate, data are shared and graphically represented in clear figures.

I have attached an annotated pdf containing some very minor corrections and requests for clarification.

Experimental design

The research question is well defined and the predictions are stated. The research gap is clear. The authors adopt a standard expectancy violation paradigm that has been used across a range of species to explore cross-modal individual recognition. The manuscript is well written and the method is explained in sufficient detail to be replicated.

Validity of the findings

The data are provided and the analysis appears well thought out and sound. There is some variation in the duration of HRV measurement and time between trials that may have introduced some noise into the data, however, conducting controlled experiments under such conditions presents understandable challenges.

The conclusions are suitably tentative given that the differences between conditions were limited and not in the expected direction. The implications of the findings are not over-stated.

Cite this review as

---

## Round 0.2 · Minor Revisions

Thank you for a very thorough and thoughtful response to the previous round of reviews. The reviewers and I appreciated your willingness to engage in the necessary revisions. Once again, I appreciate your detailed description of your analytic approach. Reviewer 3 is now satisfied with the revised MS but the other two reviewers point to some additional minor changes/considerations that would benefit the MS. I agree that you should adjust the last several sentences of your abstract and the final paragraph of the discussion so that you do not focus on implications of findings you did not obtain. For that reason, I'd like to invite you to engage in an additional minor revision to give you the opportunity to respond to their comments. I have a few additional comments of my own:

Avoid using "our" and "us" (lines 107-108)
If I understand correctly that you no longer include a reliability check on the coding that was used for analyses, you will need to add this for at least a small subset of the data (at least 10% of trials please; ideally 20%). I am sorry to have to insist on this as I recognize how time consuming coding can be, but it is extremely important to have a measure of reliability for the reported variables.
Line 476, change "which" to "that"

·

Basic reporting

Very minor point here - in the abstract (41), you talk about behavioural changes first then physiological. Then you give the physiological result before the behavioural, so I would just suggest switching it so it goes "we expected goats to show changes in physiological parameters and to respond faster and longer", to make it easier on your reader.

Otherwise this is fine.

Experimental design

It’s a shame that the new analysis has lost the careful second coding as this is an important check on the data analysis. I’m aware that the previous coding had very high agreement but would it be possible to include the second coding of a few videos to demonstrate it for the new analysis too? I realise this is a considerable amount of work but it is best practice and I am confident it will result in a similarly laudable high agreement.

Minor point -
Previously I stated: Materials and Methods Section 2.7., Lines 298-302: In the paragraph starting line 272, do you mean randomised or was this pseudo-random with counterbalancing so that you had relatively equal numbers from each group at each presentation position (1-4)? If not, how did you counteract the possible clumping of trials?
Apologies for confusing you. I meant if you have 2 congruent trials of face & voice and 2 incongruent face & voice trials, did you have these so that each goat experienced congruent 1, incongruent 1, congruent 2, incongruent 2, or did you counterbalance presentation so that an equal number had Inc1, Con1, Inc2, Con2; Con2, Con1, Inc2, Inc1, etc.? Truly random presentation might end with 10 goats getting Inc1, Con1, Inc2, Con2, and 2 getting Con1, Inc2, Inc1, Con2 etc.

Validity of the findings

The findings are valid overall but I think you're overstating the findings in the abstract and discussion, a left-over from when the results looked more positive. In both the abstract & Lines 511 & 592 seem to overstate what you’ve found. Despite a robust approach, you find no robust evidence that the goats are recognising individuals, with no significant difference in their behaviour, and that's fine as you did a thorough job of testing them with photographs. Just say they didn't which opens up questions about why not, instead of "if ... provided...." (in the abstract) when the evidence is so weak here. The heart-rate results are very hedged with caveats so I’m cautious of placing too much weight on them.

I understand the desire to say you have a positive result but at the moment there are no significant differences in the behaviours measured and the final conclusion seems to argue that you have at least preliminary evidence that goats do perform crossmodal recognition. I suspect if this was re-run with live humans, instead of photographs, you would find the evidence you need for your conclusions and that it is likely you are already planning this as a follow-up study. I look forward to seeing those results if so.

Additional comments

First I would like to thank the authors for their absolutely excellent and very thorough response to their reviews. The care which they took is a model for the system. I am very happy with the new manuscript and the changes made. The paper now reads really well and I have only minor points above to address before I feel it is fully ready. The final conclusions seem to be leftover from previous drafts but this is very understandable after so many changes.

The authors have produced a really interesting, robustly performed experiment with an intriguing result. Well done.

Cite this review as

Reviewer 2 ·

Basic reporting

In general, I found the reporting clear and easy to understand. One comment below:
Line 304-305: Goats listened to a single sequence of two voice playbacks each followed by 10s of silence in which, their behavioural and cardiac responses were measured.
You state in line 317-318 that you coded behaviour in the 10s from the start of the playback, not within the 10s silence gap. You will need to change this sentence to make sure you are telling a consistent story. Also the placement of the comma after ‘which’ is a bit confusing.

Experimental design

The experimental design appears sound however I question the addition of the HR/ HVR data from goats that were already looking at the photograph when you presented the voice recording to them. If the goats were looking at the photograph while the voice recording was being played they presumably saw that the face in the photograph was static (i.e., wasn’t in fact moving in time with the vocalisation). This could have prevented the goats from associating the face and voice with each other (even if the pairing in terms of the identity of the speaker was congruent) due to the mismatch between the dynamic auditory and static visual stimuli. This makes your data from these animals hard to interpret and arguably their exposure to stimuli is different compared to those goats that only looked at the photographs after hearing the voice stimulus.
Note that in other crossmodal studies that used photographs (e.g., Adachi et al., 2007, with dogs or Takagi et al., 2019, with cats) this simultaneous pairing of static with dynamic information was avoided by presenting the stimuli sequentially with voice followed by photograph presentation. This means that the animals would not have been exposed to the paradoxical non-moving face and voice pairings.
Therefore I would remove the data from the goats that were already looking, from your analysis.

Validity of the findings

This is a tentative finding but it is interesting and should lead to a further investigation.

Additional comments

I would like to thank the authors for taking the considerable time and effort to address my comments, the comments of other reviewers, and the editor, so thoroughly. I feel the paper is much improved.

Cite this review as

·

Basic reporting

Fine. The authors could consider shortening the final part of the abstract that goes into detail as to what the results might show, instead making it clear that more research is needed to draw any firm conclusions.

Experimental design

Fine.

Validity of the findings

The authors have been very careful with the interpretation of their findings.

Additional comments

The papers is improved and the authors have addressed all of my previous comments to my satisfaction.

Cite this review as

---

## Round 0.3 · accepted · Accept

Thank you for your revision. I think the paper now makes a nice contribution to an interesting area of the literature.